# Non-Hermitian Chiral Magnetic Effect in Equilibrium

**Maxim N. Chernodub** [1,2,*,†] and **Alberto Cortijo** [3,4,†]

1   Institut Denis Poisson UMR 7013, Université de Tours, 37200 Tours, France
2   Pacific Quantum Center, Far Eastern Federal University, Sukhanova 8, 690950 Vladivostok, Russia
3   Instituto de Ciencia de Materiales de Madrid, CSIC, Cantoblanco, 28049 Madrid, Spain
4   Departamento de Fisica de la Materia Condensada, Universidad Autonoma de Madrid, 28049 Madrid, Spain
*   Correspondence: maxim.chernodub@idpoisson.fr
†   These authors contributed equally to this work.

**Abstract:** We analyze the chiral magnetic effect for non-Hermitian fermionic systems using the bi-orthogonal formulation of quantum mechanics. In contrast to the Hermitian counterparts, we show that the chiral magnetic effect takes place in equilibrium when a non-Hermitian system is considered. The key observation is that for non-Hermitian charged systems, there is no strict charge conservation as understood in Hermitian systems, so the Bloch theorem preventing currents in the thermodynamic limit and in equilibrium does not apply.

**Keywords:** non-hermitian hamiltonian; chiral magnetic effect; equilibrium transport; Bloch theorem

## 1. Introduction

The Chiral Magnetic Effect (CME) is the generation of an electric current $\boldsymbol{J}$ in the presence of an external magnetic field $\boldsymbol{B}$ [1]:

$$\langle \boldsymbol{J} \rangle = \frac{e^2}{2\pi^2} \mu_5 \boldsymbol{B}. \tag{1}$$

The current (1) appears naturally in a particular set of physical systems characterized by a broken invariance under the spatial reflection $\mathcal{P}$. The broken $\mathcal{P}$-invariance may be realized, for example, in condensed matter systems with massless fermionic quasiparticles of a Weyl or Dirac type. Such fermions are characterized by different (left and right) chiralities, which are often said to be ascribed to different "two Weyl nodes". If the numbers of fermions with different chiralities are not equal to each other, then the system is $\mathcal{P}$-broken. The chiral imbalance is convenient to characterize by a difference denoted by $\mu_5 = \mu_L - \mu_R$ between the chemical potentials in the right ($\mu_R$) and left ($\mu_L$) Weyl nodes. The difference in the chemical potentials determines the magnitude of the CME current in Equation (1).

The CME is a relevant transport phenomenon that has its roots in the physics of quantum anomalies. The theory is said to be anomalous if there exists a quantity that is conserved at the classical level and that fails to do so when going to the quantum realm. In particular, the CME stems from the axial anomaly, which leads to non-conservation of the chiral current in Weyl systems described by the Weyl Hamiltonian of a massless particle with the wavefunction $\psi_s$:

$$H_s = s v_F \psi_s^+ \boldsymbol{\sigma} \cdot \boldsymbol{k} \psi_s. \tag{2}$$

Here, the parameter $s = \pm 1$ denotes the chirality of the particle propagating with the velocity $v_F$ and momentum $\boldsymbol{k}$ and $\boldsymbol{\sigma}$ is the vector of the Pauli matrices. In relativistic systems, Lorentz invariance forces $v_F = c$, while in condensed matter systems, the velocity $v_F$ is not constrained to any particular value. It is often said that the parameter $s = \pm 1$ labels two distinct "Weyl points".

The Weyl systems (2) possess two quantities that are conserved at the level of classical equations of motion. These are the electric charge $Q$ and the axial charge $Q_5$ that are described, respectively, by electric four-current $J^\mu$ and the chiral current ($j$ for another current to be defined below (12)) $j_5^\mu$:

$$J^\mu \equiv (Q, \boldsymbol{J}) = \bar{\Psi}\gamma^\mu\Psi, \qquad j_5^\mu \equiv (q_5, \boldsymbol{j}_5) = \bar{\Psi}\gamma_5\gamma^\mu\Psi. \tag{3}$$

Mathematically, the conservation implies that the four-divergence of the both currents is identically zero, $\partial_\mu J^\mu = \partial j_5^\mu = 0$, provided the wave function $\Psi = (\psi_R, \psi_L)^T$ satisfies the classical equation of motion $H\Psi = 0$ where the Hamiltonian $H = \mathrm{diag}\,(H_R, H_L)$ incorporates both chiral modes (2). We use the conventional nomenclature of $\gamma$ matrices in the chiral basis:

$$\gamma^0 = \begin{pmatrix} 0 & \mathbb{1} \\ \mathbb{1} & 0 \end{pmatrix}, \qquad \boldsymbol{\gamma} = \begin{pmatrix} 0 & \boldsymbol{\sigma} \\ -\boldsymbol{\sigma} & 0 \end{pmatrix}, \qquad \gamma^5 = \begin{pmatrix} -\mathbb{1} & 0 \\ 0 & \mathbb{1} \end{pmatrix}. \tag{4}$$

In the presence of conventional electromagnetic fields, the quantum fluctuations lead to nonconservation of the chiral current. Technically, the loss of conservation appears as a result of the so-called triangle diagrams of virtual fermions that lead to [2,3]:

$$\partial_\mu j_5^\mu = \frac{1}{2\pi^2}\boldsymbol{E} \cdot \boldsymbol{B}. \tag{5}$$

The triangle diagrams that give rise to the non-conserved current (5) are also responsible for the generation of the current in Equation (1). The form of the anomaly can be partially fixed by some algebraic constraints on an effective action of the theory that leads the right-hand side of Equation (5). These constraints are imposed by the algebraic structure of the symmetry that is anomalously broken [4]. Technically, the relation of the chiral (triangular) anomaly to the generation CME current (1) requires a rigorous derivation that takes into account the Wess–Zumino consistency conditions [5]. In the presence of the chiral gauge fields $A_\mu^5$ that are coupled to the chiral current $j_5^\mu$, the anomalous effects become more subtle and the currents (3) have to be modified consistently. In our work, $A_\mu^5 \equiv 0$, so that we will use the straightforward definition (3) for vector and axial currents.

In general, two ways have been proposed to create an environment that is able to generate the anomalous current (1). The first approach is to drive the system out of equilibrium in order to reach a stationary regime where $\mu_5 \neq 0$. This regime may be achieved by applying simultaneously an electric field $\boldsymbol{E}$ parallel to $\boldsymbol{B}$ so that the chiral anomaly creates a charge imbalance via the chiral anomaly (5) and generates a nonzero chiral chemical potential $\mu_5$. Then, the system generates a non-equilibrium electric current via the CME mechanism (1). Notice that the chiral imbalance $\mu_5 \neq 0$ does not exist in an equilibrium regime as the populations of the left-hand particles and the right-hand particles mix with each other due to interactions and then relaxes towards $\mu_5 = 0$ equilibrium. Therefore, the anomalous electric current (1) is zero in thermal equilibrium.

The second approach could consist of moving the position of the Weyl nodes in energies without carrying the system out of equilibrium:

$$\langle \boldsymbol{J} \rangle = \frac{e^2}{4\pi^2}(\epsilon_R - \epsilon_L)\boldsymbol{B} = 0, \tag{6}$$

where the energies $\epsilon_{L,R}$ are the position of left and right Weyl points in energy. The equilibrium current (6) is, however, vanishing in Hermitian systems. Physically, the current is zero because the difference in the energies of the right- and left-hand chiral fermions does not create the true chiral imbalance. Mathematically, the current (6) does not exist because the difference in energies $(\epsilon_R - \epsilon_L)$ is sensitive to the chirality of the fermion and, therefore, is nothing but the zeroth component of a chiral gauge field, $A_0^5$. Once the chiral gauge field appears, the definition of the physical electric current starts to differ from the naive covariant version (3) by an addition of an extra term coming from the so-called Bardeen polynomials. This term cancels out this energy difference precisely, and the physical

(so-called "consistent") version of the current vanishes in thermal equilibrium (6). For further details and technicalities, we refer the interested reader to [5].

Despite that in thermodynamic equilibrium, the axial chemical potential is zero, $\mu_5 = 0$, the vector chemical $\mu = \mu_R + \mu_L$ for a generic fermionic system may still be nonzero. In the presence of the background magnetic field $\boldsymbol{B}$, the system generates (via the same chiral anomaly) the chiral current:

$$\langle J_5 \rangle = \frac{e^2}{2\pi^2} \mu \boldsymbol{B}, \tag{7}$$

which is the direct analogue of the CME (1), but now the chiral sector. Equation (7) describes the Chiral Separation Effect (CSE), which will play a role in our derivations below along with the CME.

While the non-equilibrium situation has been explored extensively in the literature leading, for instance, to the celebrated negative quadratic magnetoresistivity in Weyl metals, the equilibrium scenario appears not to be possible, and to date, there is the consensus that the CME is not possible in equilibrium [6–8].

The statement of the absence of CME in equilibrium can be seen as an extension of a no-go theorem given by Bloch, concerning the existence of equilibrium currents in solids in the thermodynamic limit [9]. This theorem has been extended to chiral matter in [7] and refined in [8] (the absence of CME in equilibrium using the chiral kinetic formalism was obtained in [6]). There are three elements usually associated with this theorem in chiral matter: the existence of Weyl nodes that always come in pairs [10,11], (local) gauge invariance, and of course, the assumption that the system is in the equilibrium state. As we have mentioned, it is known how to break the second condition and drive the system out of equilibrium. Recently, it has been proposed that the first assumption of having pairs of Weyl nodes can be broken in Weyl superconductors, where an external magnetic field induces a gap in one of the Weyl nodes (and its particle-hole conjugate), leaving effectively a single Weyl node [12]. However, we stress here that the presence of Weyl nodes is not a strict requirement for the absence (or presence) of CME [7,8].

Interestingly, non-Hermitian fermionic systems appear to be a promising physical environment that can be realized in real experiments. Currently, there is a surge of interest in non-Hermitian systems for many different reasons, ranging from very fundamental questions in the quantum (and statistical) theory of fields and the role of topology in non-Hermitian systems [13–15], to applied science. Among them, especially interesting are the non-Hermitian systems that display a real spectrum, as the $\mathcal{P} * \mathcal{T}$-symmetric systems or the quasi-Hermitian systems. Although non-Hermitian, they display a unitary evolution, and it is possible to define a consistent thermodynamics for them [16].

## 2. The Model

We will study a non-Hermitian extension of the massive Dirac model in $(3+1)$ dimensions, where, together with the usual mass term $m$, an anti-Hermitian mass $m_5$ is introduced [17–19]:

$$H = \boldsymbol{\alpha} \cdot \boldsymbol{k} + m\hat{\beta} + m_5 \hat{\beta}\gamma_5. \tag{8}$$

Here, we used the original Dirac notations $\boldsymbol{\alpha} \equiv \gamma^0 \boldsymbol{\gamma}$ and $\hat{\beta} \equiv \gamma^0$, where the Dirac gamma matrices are given in Equation (4) and $\boldsymbol{k}$ describes the momentum of the particle. The advantage of this model is that the first two terms of the right-hand side of Equation (8) are Hermitian by themselves, so the only non-Hermitian (anti-Hermitian) term is $m_5\gamma_5$. When $m_5$, the model Equation (8) corresponds to the usual Dirac model for relativistic fermions. Furthermore, it constitutes the low energy model for the bulk states of topological insulators, and when $m = 0$ or $m = m(\boldsymbol{k})$ becomes a function with nodal points in momentum space, this model describes Weyl fermions [20].

To date, there is no known experimental realization of an electronic system with the non-Hermitian mass term $m_5\hat{\beta}\gamma_5$. However, we are not aware of any no-go theorem that would forbid this term from appearing in open systems. Therefore, we consider the model (8) as a generic system that captures the

essential properties of the non-Hermitian mass. Our aim is to show, conceptually, that the equilibrium chiral magnetic effect is, in principle, possible in a generic non-Hermitian system.

It has already been stated in the literature that non-Hermitian Hamiltonians are not gauge-invariant in general. This can be viewed as the fact that the Noether theorem relating continuous symmetries and conserved currents in field theories does not hold in non-Hermitian systems [18,19,21]. For this reason, there is some arbitrariness when defining a coupling to electromagnetic fields in the Hamiltonian (8). In the present work, we are interested in comparing our results with the ones in Hermitian systems, so we will consider the coupling of electromagnetic fields to the model (8) with $m_5 = 0$, which is Hermitian (and the principle of local gauge invariance holds), and later, we switch on the non-Hermitian term $m_5 \hat{\beta} \gamma_5$.

Furthermore, that we cannot apply the Noether theorem for Hermitian systems to non-Hermitian ones does not imply that conserved currents exist for the latter.

In a conventional Hermitian quantum mechanics, the time dependence of any operator $\mathcal{O}$ can be determined in the Heisenberg picture:

$$\mathcal{O}(t) = e^{iH^+ t} \mathcal{O} e^{-iHt}. \tag{9}$$

If we naively try to proceed further following the same steps for a non-Hermitian (NH)system, we will get an unconventional expression for the time variation of the operator $\mathcal{O}(t)$[22,23]:

$$\dot{\mathcal{O}} \equiv \frac{d\mathcal{O}(t)}{dt} = i e^{iH^+ t} \left( H^+ \mathcal{O} - \mathcal{O}H \right) e^{-iHt}. \tag{10}$$

For Hermitian systems, $H^+ = H$, and we recognize the commutation with $H$ as the condition any operator must satisfy to be a conserved quantity. For non-Hermitian systems, we immediately see that an operator is a conserved quantity if, instead of commuting with the Hamiltonian, it fulfills the quasi-Hermiticity condition: $H^+ \mathcal{O} = \mathcal{O}H$, so $\dot{\mathcal{O}} = 0$. In the case of the $U(1)$ charge symmetry, it is clear that the generator of this symmetry in Hermitian systems commutes with $H$, but does not satisfy the quasi-Hermiticity condition, so it is not a conserved quantity for non-Hermitian systems. As we will see in the next paragraphs, it is possible to find operators that, not commuting with $H$, satisfy the quasi-Hermiticity condition, thus defining conserved quantities.

Before computing the CME and CSE for the system at hand, it is convenient to understand what is the physical meaning of the conserved currents associated with the Hamiltonian (8). The most convenient way is the following [24]: The Hamiltonian (8) is a quasi-Hermitian Hamiltonian that satisfies the relation $H\eta = \eta H^+$, where $\eta$ is some positive definite operator called the metric operator. The condition of the metric operator of being positive definite allows us, among other things, to define a non-unitary similarity transformation $S$ ($\eta = S^+ S$) that maps the NH Hamiltonian $H$ in Equation (8) onto a Hermitian Hamiltonian $\hat{H}$ (for further details, we refer to Appendix B). Then, we can find the conserved currents in the auxiliary Hermitian Hamiltonian and use the mapping $S$ on these conserved currents to find the corresponding conserved quantities in the non-Hermitian model.

In certain systems, where the similarity matrix $S$ cannot be constructed explicitly, it is still possible to identify certain conserved quantities. These quantities are associated with the operators that are symmetries of the system. Namely, given an operator $\mathcal{O}$, we can construct another operator $\mathcal{O}' = \eta \mathcal{O}$, whose time evolution is described, according to Equations (9) and (10), as follows:

$$\frac{d\mathcal{O}'(t)}{dt} = i\eta e^{iHt}[H, \mathcal{O}]e^{-iHt}. \tag{11}$$

We see that this new operator $\mathcal{O}'$ now possesses a conventional time evolution of the quantum mechanics in the Heisenberg picture. Furthermore, if the original operator $\mathcal{O}$ is a symmetry of the problem (that is $[H, \mathcal{O}] = 0$), then the new operator $\mathcal{O}' = \eta \mathcal{O}$ defines a conserved quantity as well. This discussion allows us to motivate the use of a bi-orthogonal formulation in our paper. It is clear,

indeed, that the expression in Equation (11), constructed with the help of the operator $\mathcal{O}' = \eta\mathcal{O}$, follows the standard time evolution in terms of the conjugate wavefunctions ($\langle\psi|, |\psi\rangle$).

Alternatively, instead of using the modified operator $\mathcal{O}'$, we could have perfectly maintained the operator $\mathcal{O}$ and defined a modified conjugate wavefunction $\langle\psi|\eta$. The pair ($\langle\psi|\eta, |\psi\rangle$) is called a bi-orthogonal set. We will make use of this formulation in the next sections.

Let us now apply the mentioned results to the model defined in Equation (8). The corresponding procedure, developed in [19], utilizes the metric operator $\eta = \mathbf{1} + \frac{m_5}{m}\gamma_5$. It turns out that the Hermitian model associated with the non-Hermitian Hamiltonian $H$ corresponds to a massive Dirac spinor $\chi$ with mass $M = \sqrt{m^2 - m_5^2}$. We construct the $U(1)$ conserved current $\hat{j}^\mu = \bar{\chi}\gamma^\mu\chi$ associated with the spinor $\chi$. After using the inverted mapping $S$, we get the corresponding current in the non-Hermitian system in terms of the field $\psi^+$:

$$j^\mu = \psi^+\gamma^0(\mathbf{1} + \frac{m_5}{m}\gamma_5)\gamma^\mu\psi = \psi^+\gamma^0\eta\gamma^\mu\psi = \bar{\psi}\eta\gamma^\mu\psi. \tag{12}$$

Since $\partial_\mu\hat{j}^\mu = 0$, we can trivially show that the current $j^\mu$ corresponds to a conserved quantity $\partial_\mu j^\mu = 0$. We thus see that the current $j^\mu = \eta J^\mu$ is a conserved current with $J^\mu = \gamma^\mu$ being a symmetry of the Hamiltonian in Equation (8).

As will be discussed in the next section, the most important consequence of having the conserved current $j^\mu$ is that we can define a chemical potential $\mu$ associated with $j^0 = \eta$.

We immediately see that the current is made of a piece proportional to the identity, as it corresponds to an abelian current in the Hermitian case, together with a chiral current weighted by $m_5/m$ that implies a chiral imbalance. We will show in the next section and in Appendix B that a system defined by the Hamiltonian (8) that exchanges particles in a manner defined by this precise chemical potential $\mu$ defines it in a truly equilibrium thermal state with non-vanishing CME and CSE.

## 3. Computation of CSE and CME with Biorthogonal Quantum Mechanics

Here, we will tackle the problem using the biorthogonal quantum mechanics formalism [25]. Within this formalism, we distinguish between the eigenstates of $H$: $H\psi_s = \varepsilon_k^s\psi_s$, their complex conjugates: $\psi_s^+H^+ = \psi_s^+\varepsilon_k^s$, the bi-orthogonal states $\phi_s$: $H^+\phi_s = \varepsilon_k^s\phi_s$, and their complex conjugates, $\phi_s^+H = \varepsilon_k^s\phi_s^+$. The point is that, because $H$ is not Hermitian, $\psi_s \neq \phi_s$ and $\psi_s^+ \neq \phi_s^+$. Furthermore, for the same lack of Hermiticity, the states are not orthogonal $\langle\psi_s^+|\psi_{s'}\rangle \neq \delta_{ss'}$, where $\langle\cdot|\cdot\rangle$ is the standard scalar product in the corresponding Hilbert space. However, the state sets $\psi_s$ and $\phi_s$ form a bi-orthogonal basis:

$$\langle\psi_s^+|\phi_{s'}\rangle = \langle\phi_s^+|\psi_{s'}\rangle \propto \delta_{ss'}. \tag{13}$$

For the model (8), we can define a metric operator $\eta$, that not only fulfills the quasi-Hermiticity condition, $\eta H = H^+\eta$, but is positive definite. The existence of such an operator simplifies the construction of the bi-orthogonal basis sets, since these two bases are related to each other through $\eta$:

$$\phi_s = \frac{1}{\langle\psi_s^+|\eta\psi_s\rangle}\eta\psi_s. \tag{14}$$

With this particular normalization, we have $\langle\psi_s^+|\phi_{s'}\rangle = \langle\phi_s^+|\psi_{s'}\rangle = \delta_{ss'}$. For the Hamiltonian at hand (8), such a metric operator $\eta$ is $\eta = \mathbf{1} + \frac{m_5}{m}\gamma_5$ [19]. The existence of a metric operator allowing us to define a well-defined inner product in the corresponding Hilbert space, defines an unitary time evolution of the states, as long as the spectrum is real, so a consistent description of quantum mechanics is allowed for the system, although being non-Hermitian. Furthermore, it is now easy to see that any time operator will evolve using the conventional Heisenberg picture within the bi-orthogonal formalism.

Another relevant consequence of the existence of the metric operator is that $\eta$ is a conserved quantity, since, as we mentioned, the matrix $\eta$ fulfills the pseudo-Hermiticity condition (remember Equation (10)). Although $\eta$ does not commute with the non-Hermitian Hamiltonian $H$ [26], it allows for a construction of an unitary evolution. The existence of a conserved quantity makes it possible to define a Lagrange multiplier $\mu$ associated with the operator $\eta$. Since $\eta$ is a conserved quantity, that Lagrange multiplier $\mu$ plays the role of the chemical potential. Consequently, we can define a new Hamiltonian:

$$\mathcal{H} = H - \mu\eta, \tag{15}$$

as is done in the standard Hermitian statistical mechanics. Of course, due to the non-Hermitian nature of the problem, the conserved quantity does not need to commute with $H$. Instead, to be conserved, the corresponding operator should satisfy the aforementioned pseudo-Hermiticity condition.

However, the existence of a common basis between $H$ and any operator $\mathcal{O}$ is possible if and only if the operator $\mathcal{O}$ commutes with the Hamiltonian $H$, irrespective of the Hermiticity of $H$. This fact means that we will not be able to find a common basis for $\eta$ and $H$ in terms of the eigenstates of the number operator, as happens in conventional Hermitian quantum mechanics. This problem may be circumvented by building the bi-orthogonal basis, which is, in turn, constructed by diagonalizing the new Hamiltonian (15) $\mathcal{H}$ instead of the original Hamiltonian $H$.

In order to compute the non-Hermitian version of the chiral magnetic effect, we consider the model Equation (15) in a classical background of an external constant magnetic field $\boldsymbol{B}$ that points along the third dimension. As it is a trivial exercise to obtain the Landau levels for this model, we do not present the details of the derivation. However, we are willing to highlight two properties of these Landau levels.

First, the algebraic wavefunction structure in the non-Hermitian system does not differ from the Hermitian case: The system is translationally invariant along the direction of the magnetic field $\boldsymbol{B}$, so that the momentum along this direction is a conserved integral of motion. Thus, we may use a standard Fourier transformation of the wavefunction along this direction. The dispersion relation of $\mathcal{H}$ for a non-zero chemical potential is presented in Figure 1a.

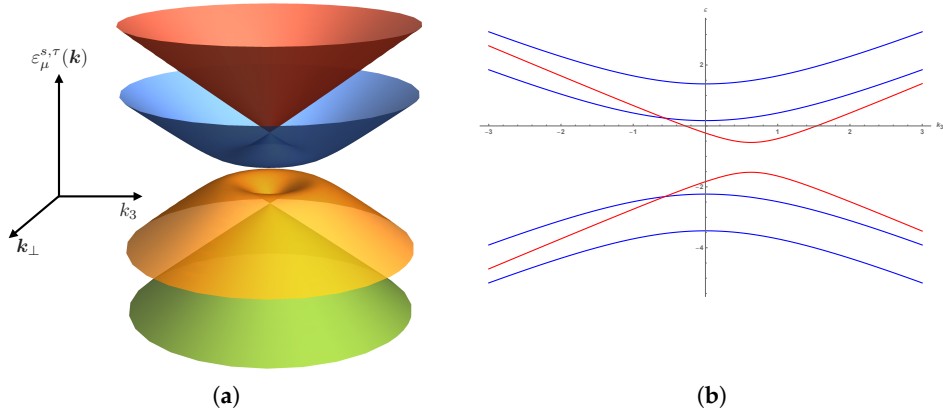

(**a**)                                                                         (**b**)

**Figure 1.** (Color online) (**a**) Band structure of $\mathcal{H}$ at zero magnetic field, but finite chemical potential. Contrary to Hermitian systems, the presence of chemical potentials might modify the spectrum strongly. (**b**) Landau level spectrum for the non-Hermitian model for finite chemical potential $\mu$. Finite values of $\mu$ shift the Lowest Landau Level (LLL) spectrum (red) not only upwards or downwards, but also laterally. It is the lateral shift that makes the nonvanishing contribution from the LLL to the Chiral Magnetic Effect (CME).

Second, the wavefunctions are highly degenerate as in the Hermitian case, with the same Landau degeneracy. The current along the magnetic field becomes diagonal. Therefore, following a standard procedure used in the Hermitian case, we can integrate over the transverse spatial directions when

computing quantum averages. This approach greatly simplifies the calculations and allows us to consider the problem as quasi-one-dimensional.

Although from the perspective of constructing a thermal equilibrium ensemble, the lack of Hermiticity in the system might be supposed as a problem [27], here, we will argue that this is not actually the case in our model system under general grounds, due to the requirement that the one-particle correlation function built from the bi-orthogonal basis satisfies the Kubo–Martin–Schwinger (KMS) periodicity condition: $\langle \Psi^+(0)\Psi(\tau')\rangle = -\langle \Psi^+(\beta)\Psi(\tau')\rangle$ [28–30]. It is known in the context of quantum statistical mechanics that states satisfying the KMS boundary condition extremize the von Neumann entropy $S = -Tr[\rho \log \rho]$, $\rho$ being the density matrix operator, so they describe equilibrium states [30]. The point is to notice that, for non-zero $\mu$, the time evolution of any field operator is done through the exponential of $\mathcal{H}$ (and not of $H$), so then, the one-particle correlation function will satisfy the KMS boundary condition, and we will be able to build an equilibrium ensemble [31]. In Appendix B, we provide an explicit proof that the correlation functions of the non-Hermitian system considered in the present work can be mapped to the correlation functions of an equilibrium thermal state, thus satisfying the KMS condition. Furthermore, this fact has been pointed out in the existing literature of non-Hermitian systems [32,33].

In our particular case, using the effective one-dimensional model, we will focus only on the Lowest Landau Level (LLL) after integrating over the perpendicular coordinates, and explicitly writing the Landau level degeneracy $\rho = 2\pi e B_3$, the equilibrium thermal average of any observable $\mathcal{O}$ will be:

$$\langle \mathcal{O} \rangle = \frac{e^2 B_3}{4\pi^2} \sum_{\omega_n} \int_{-\infty}^{\infty} dk_3 \, Tr[\mathcal{O} G_0(i\omega_n, k_3)], \tag{16}$$

where $G_0(i\omega_n, k_3)$ is the single-particle propagator in imaginary time:

$$G_0(i\omega_n, k_3) = \sum_{s,n} \frac{|\psi_s\rangle \langle \phi_s|}{i\omega_n - \varepsilon_\mu^s(k_3)}, \tag{17}$$

and $(\psi_s, \phi_s)$ are the bi-orthogonal sets of single-particle eigenstates of the model (15) in the presence of an external magnetic field $\boldsymbol{B} = B_3 \hat{z}$: $\mathcal{H}\psi_s = \varepsilon_\mu^s(k_3)\psi_s$, $\mathcal{H}^+\phi_s = \varepsilon_\mu^s(k_3)\phi_s$. The generic label $s$ comprises band labeling, the spin index $\tau$, and the Landau level index $N$.

For the operators defined as $\mathcal{O} = \frac{\partial \mathcal{H}}{\partial \lambda}$, we can generalize the Feynman–Hellmann theorem to the bi-orthogonal basis (See Appendix A), if the eigenstates are real:

$$\langle \phi_s^+ | \mathcal{O}\psi_s \rangle = \left\langle \phi_s^+ | \frac{\partial \mathcal{H}}{\partial \lambda}\psi_s \right\rangle = \frac{\partial \varepsilon_s}{\partial \lambda}, \tag{18}$$

obtaining, after performing the Matsubara summation,

$$\langle \mathcal{O} \rangle = \frac{e^2 B_3}{4\pi^2} \sum_{s,n} \int_{-\infty}^{\infty} dk_3 \frac{\partial \varepsilon_\mu^s(k_3)}{\partial \lambda} n_F(\varepsilon_\mu^s(k_3)), \tag{19}$$

where $n_F(x)$ is the Fermi distribution function in absence of the chemical potential. The chemical potential is part of the spectrum.

For the case of CME, $J_3 = \frac{\partial \varepsilon_\mu^s(k_3)}{\partial k_3}$, so:

$$\left\langle J^3 \right\rangle = \frac{e^2 B_3}{4\pi^2} \sum_{s,N} \int_{-\infty}^{\infty} dk_3 \frac{\partial \varepsilon_\mu^{s,N}(k_3)}{\partial k_3} n_F(\varepsilon_\mu^{s,N}(k_3)). \tag{20}$$

The dispersion relation for the LLL ($N = 0$) sector is (see Figure 1b):

$$\varepsilon_\mu^{s,0}(k_3) = -\mu + s\sqrt{(k_3 - \delta\mu)^2 + m^2(1 - \delta^2)}, \tag{21}$$

where $\delta = \frac{m_5}{m}$ and $s = \pm 1$, while for $N > 0$, we have:

$$\varepsilon_\mu^{s,\tau,N}(k_3) = -\mu + s\sqrt{(\sqrt{k_3^2 + \omega_c^2 N} + \tau\delta\mu)^2 + m^2(1 - \delta^2)}. \tag{22}$$

For the $N > 0$ Landau levels, the spin degree of freedom $\tau = \pm 1$ appears explicitly. In Figure 1b, we plot the Landau level spectrum for $N = 0$ and $N > 0$. The all-important difference between the eigenenergies for $N = 0$ and $N > 0$ is that, while $\varepsilon_\mu^{s,\tau,n}(k_3)$ with $N > 0$ is an even function of $k_3$ for any value of $m, \delta = m_5/m$ and $\mu$, the energy $\varepsilon_\mu^{s,0}(k_3)$ with $N = 0$ is not. That means that, when taking the derivative with respect to $k_3$ and integrating over a symmetric interval, the $N > 0$ Landau levels will not contribute to the integral in (20), but the $N = 0$ will.

The result turns out to be (see Figure 2):

$$\left\langle J^3 \right\rangle = \frac{e^2 B_3}{2\pi^2} \frac{m_5}{m} \mu. \tag{23}$$

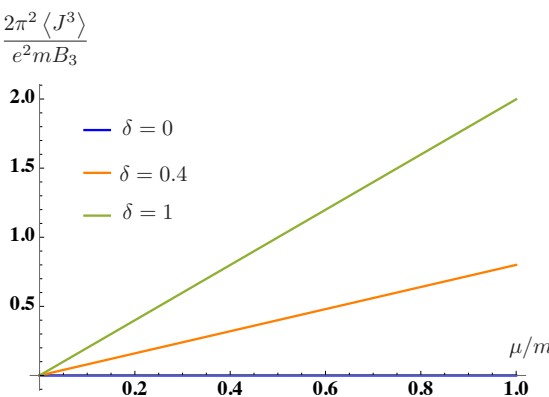

**Figure 2.** (Color online) CME as a function of $\mu/m$ for three values of $\delta = m_5/m$. The vanishing CME for the Hermitian case, $\delta = 0$, is recovered.

This is the principal result of this letter. For non-zero values of the mass $m_5$, which is the parameter that controls the non-Hermiticity of $\mathcal{H}$, there is a non-vanishing CME in equilibrium.

The Chiral Separation Effect (CSE) is obtained by computing the average value of the chiral current, represented by the operator $J_5^i = e\alpha^i\gamma_5$. We can follow the same route as in the case of the CME. We will add a term $b_3\alpha_3\gamma_5$ to the Hamiltonian (15) and compute the spectrum in the presence of the parameter $b_3$. Then, we apply the Hellman–Feynman theorem to it, taking the derivative with respect to $b_3$ and constructing the expectation value for each Landau level. We send the parameter $b_3$ to zero after the calculation. It is a lengthy, but straightforward calculation to check that for the $n > 0$ sector, $\partial\varepsilon_\mu^{s,n}(k_3, b_3)/\partial b_3$ is an odd function of $k_3$ in the limit $b_3 \to 0$ for all values of $m, m_5,$ and $\mu$. This implies that the integral over $k_3$ is zero, and they do not contribute to the CSE. In contrast, for the $n = 0$ sector, we simply have $\partial\varepsilon_\mu^s(k_3, b_3)/\partial b_3 = 1$, so:

$$\left\langle J_5^3 \right\rangle = \frac{e^2 B_3}{4\pi^2} \sum_s \int_{-\infty}^{\infty} dk_3 n_F(\varepsilon_\mu^{s,0}(k_3, b_3 = 0)). \tag{24}$$

We plot $\left\langle J_5^3 \right\rangle$ in Figure 3a,b as a function of $\delta = m_5/m$ for fixed $m$, and as function of $m$ for fixed $\delta$. Performing the integral, we finally have:

$$\left\langle J_5^3 \right\rangle = \frac{e^2 B_3}{2\pi^2}\left(\frac{1}{\epsilon} + \Theta[\mu - m]\sqrt{\mu^2 - m^2(1 - \delta^2)}\right), \tag{25}$$

with $\epsilon \ll 1$. We note that there is a divergent contribution in the CSE. It is a particular feature of $(1+1)$ dimensions that there is a duality between the chiral and charge currents. The charge current operator representing the CME is the chiral density, while the chiral current operator $j_1^5$ that is relevant for the study of the CSE is the same operator as the charge density. Having the same origin as the standard charge density, we regularize it in the same way.

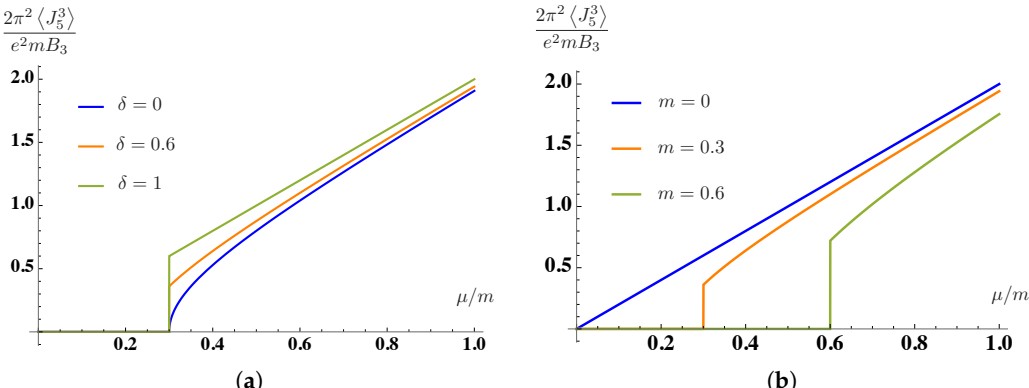

**Figure 3.** (Color online) (**a**) Regularized CSE as a function of $\mu$ for three values of $\delta = m_5/m$. We fix the mass parameter to be $m = 0.3$. (**b**) CSE as a function of $\mu$ for three values of $m$ and fixed $\delta = 0.6$.

## 4. Conclusions

In the present letter, we demonstrated that CME in equilibrium is possible when non-Hermitian systems are considered. The key ingredient is to realize that the CME is zero if charge conservation is imposed in the system. However, charge conservation, associated with the $U(1)$ symmetry, is not fulfilled in non-Hermitian systems, as it is done in conventional Hermitian ones.

Another fact to pay attention to is that there is no unique metric operator associated with the non-Hermitian Hamiltonian fulfilling the pseudo-Hermiticity condition. While there is no practical consequence of this regarding the construction of a bi-orthogonal basis (the average of observables does not depend on any particular choice of the metric operator), this observation is relevant as we can associate different chemical potentials with different metric operators understood as conserved quantities in the non-Hermitian sense. Interestingly, all the metric operators are related to each other by a similarity transformation [24], so we can generalize the results obtained here to other chemical potentials by modifying the spectrum correspondingly.

Finally, to the best of our knowledge, there are no experimental realizations of fermionic non-Hermitian systems with the real spectrum to test our predictions. However, there are impressive experimental advances in the area of non-Hermitian $\mathcal{PT}$-symmetric photonic systems and other condensed matter analogs [34,35]. In fact, the experimental observation of the CME employing superconducting quantum circuit technology and synthetic magnetic fields has been recently proposed [36]. We suggest the same experimental setup to test our theory, by extending the experimental setup with equal gain-loss [37]. Besides, other topological equilibrium effects similar to the CME have been proposed to occur in electromagnetism [38–41], the optical helicity and the optical chirality being the electromagnetic symmetries that play the role of the chiral symmetry in ultrarelativistic fermionic systems. There, the biorthogonal formalism has been probed to be useful to handle the effect of dissipation and loss in electromagnetism [42,43]. The natural question is then to see how the topologically-related responses associated with these symmetries are modified by the presence of non-Hermitian effects.

**Author Contributions:** All authors contributed equally to this work. All authors have read and agreed to the published version of the manuscript.

**Funding:** This research was funded by MINECO/AEI/FEDER, UE Grant No. FIS2015-73454-JIN., the Comunidad de Madrid MAD2D-CM Program (S2013/MIT-3007), the Ramon y Cajal program through the grant RYC2018-023938-I, and Grant No. 0657-2020-0015 of the Ministry of Science and Higher Education of Russia.

**Acknowledgments:** We kindly acknowledge inspiring conversations with K. Landsteiner about non-Hermitian systems and the physics underlying the CME.

**Conflicts of Interest:** The authors declare no conflict of interest.

## Appendix A. The Hellman–Feynman Theorem for Bi-Orthogonal Systems

In this Appendix, we give a proof of the extension of the Hellman–Feynman theorem to bi-orthogonal systems with the real spectrum.

As discussed in the main text, the bi-orthogonal basis is constructed with two sets of states satisfying $H|n\rangle = \varepsilon_n |n\rangle$, $H^+ |\bar{n}\rangle = \varepsilon_n |\bar{n}\rangle$ and $\langle n| H^+ = \langle n| \varepsilon_n$, $\langle \bar{n}| H = \langle \bar{n}| \varepsilon_n$, together with the normalization condition $\langle \bar{n}|n'\rangle = \langle n'|\bar{n}\rangle = \delta_{nn'}$.

Let us consider a Hamiltonian $H$ depending on some parameter $\lambda$. To ease notation, we will keep the dependence with the generic parameter $\lambda$ implicit in the eigenstates and eigenvalues. We are interested in computing the averaged value:

$$\left\langle \bar{n}| \frac{\partial H}{\partial \lambda} |n \right\rangle. \tag{A1}$$

Then, we compute:

$$\frac{\partial}{\partial \lambda} \langle \bar{n}|H|n\rangle = \frac{\partial}{\partial \lambda}(\varepsilon_n \langle \bar{n}|n\rangle) =$$
$$= \left\langle \frac{\partial}{\partial \lambda}\bar{n}|H|n \right\rangle + \left\langle \bar{n}|H|\frac{\partial}{\partial \lambda}n \right\rangle + \left\langle \bar{n}| \frac{\partial H}{\partial \lambda} |n \right\rangle =$$
$$= \varepsilon_n \frac{\partial}{\partial \lambda}(\langle \bar{n}|n\rangle) + \left\langle \bar{n}| \frac{\partial H}{\partial \lambda} |n \right\rangle. \tag{A2}$$

$|s\rangle$ and $|\bar{n}\rangle$ are eigenstates of $H$ and $H^+$ with the same eigenvalue $\varepsilon_n$. Simplifying a little, we finally have:

$$\frac{\partial \varepsilon_n}{\partial \lambda} = \left\langle \bar{n}| \frac{\partial H}{\partial \lambda} |n \right\rangle, \tag{A3}$$

which is the result we wanted to prove.

## Appendix B. Thermal Equilibrium Condition in Quasi-Hermitian Systems

For Hermitian systems, the condition of thermal equilibrium can be formally established by showing that the Hermitian system satisfies the Kubo–Martin–Schwinger (KMS) boundary condition for the imaginary-time propagator [30]. For quasi-Hermitian systems, it is possible to describe equilibrium in the same way, making use of the existing non-unitary mapping between the non-Hermitian and Hermitian Hamiltonians. In what follows, we will restrict ourselves to non-Hermitian systems described by Hamiltonian operators that do not depend on time.

Let us consider two operators $A(\tau)$ and $B(\tau)$ in the Heisenberg picture (and in the imaginary time formalism) described by the Hamiltonian $\mathcal{H}$ (we consider that the chemical potentials associated with the symmetries of the problem were already included in $\mathcal{H}$). The KSM condition can be stated as the following identity:

$$Tr[e^{-\beta \mathcal{H}} A(\tau)B(\tau')] = Tr[e^{-\beta \mathcal{H}} B(\tau')A(\tau + \beta)]. \tag{A4}$$

If $A = \psi^+$ and $B = \psi$ are field operators that anticommute, we have:

$$Tr[e^{-\beta \mathcal{H}} \psi^+(0)\psi(\tau')] = -Tr[e^{-\beta \mathcal{H}} \psi^+(\beta)\psi(\tau')]. \tag{A5}$$

As explained in [31], this means that the thermal averaged propagator $\langle T\psi^+(\tau)\psi(\tau')\rangle$ ($T$ refers to the Dyson time ordering) is an antiperiodic function of $\tau$ with period $\beta$. This allows the development of all the machinery of thermal field theory.

In order to show how this works for quasi-Hermitian systems, it is enough to show that, for a quasi-Hermitian Hamiltonian, it is possible to construct a Hermitian partner through a non-unitary mapping between them, so we map the statistical averages using the bi-orthogonal basis in the non-Hermitian case, map them to their Hermitian counterparts, establish the KSM condition in the latter, and go back to the non-Hermitian case, inverting the mentioned mapping.

As demonstrated in [32], the existence of a metric operator $\eta$ allows us to define the non-unitary mapping $S$ of some quasi-Hermitian Hamiltonian $\mathcal{H}$ to a Hermitian partner $\hat{\mathcal{H}}$, with $\hat{\mathcal{H}} = \hat{\mathcal{H}}^+$ (we will denote the Hermitian partners of operators with the hat symbol):

$$\hat{\mathcal{H}} = S\mathcal{H}S^{-1}. \tag{A6}$$

Furthermore, we can define the Hermitian partner of any operator associated with the quasi-Hermitian system in the same way:

$$\hat{\mathcal{O}} = S\mathcal{O}S^{-1}. \tag{A7}$$

This includes the field operators $\Psi$ and $\Psi^+$ in the second quantization formalism. As discussed in the main text, the existence of the metric operator $\eta$ allows us to construct a well-behaved scalar product in the Hilbert space and to construct bi-orthogonal basis sets, $|n\rangle$ and $|\bar{n}\rangle$. In this way, we can define the following statistical average (here, we will use the suffix *bi* to denote the statistical average with the bi-orthogonal basis):

$$
\begin{aligned}
\langle \mathcal{O} \rangle_{bi} &\equiv \sum_n \left\langle \phi_n^+ e^{-\beta\mathcal{H}} \mathcal{O}\psi_n \right\rangle = \sum_n \frac{1}{\langle \psi_n^+ \eta \psi_n \rangle} \left\langle \psi_n^+ \eta e^{-\beta\mathcal{H}} \mathcal{O}\psi_n \right\rangle = \\
&= \sum_n \frac{1}{\langle \hat{\psi}_n^+ \hat{\psi}_n \rangle} \left\langle \hat{\psi}_n^+ \underbrace{S^{-1}S}_{1} \underbrace{SS^{-1}}_{1} e^{-\beta\hat{\mathcal{H}}} \underbrace{SS^{-1}}_{1} \hat{\mathcal{O}} \underbrace{SS^{-1}}_{1} \hat{\psi}_n \right\rangle = \\
&= \sum_n \left\langle \hat{\psi}_n^+ e^{-\beta\hat{\mathcal{H}}} \hat{\mathcal{O}}\hat{\psi}_n \right\rangle = \langle \hat{\mathcal{O}} \rangle.
\end{aligned}
\tag{A8}
$$

In the second line, we used $\eta = SS$ ($S^+ = S$ in our particular case) and that the eigenstates of the non-Hermitian $\mathcal{H}$ are related to the eigenstates of the Hermitian partner $\hat{\mathcal{H}}$ through $\hat{\psi}_n = S\psi_n$. Furthermore, we consider that the states $\psi_n$ of the Hermitian partner are conveniently normalized: $\langle \hat{\psi}_n^+ \hat{\psi}_n \rangle = 1$.

To guarantee the proper normalization of (A8), we need to relate the partition function in the quasi-Hermitian system and its Hermitian partner. This is a particular case of the previous identity, as we can choose $\mathcal{O} = \mathbf{1}$ and obtain the equality of the corresponding partition functions:

$$
\begin{aligned}
Z_{bi} &= \sum_n \left\langle \phi_n^+ e^{-\beta\mathcal{H}} \psi_n \right\rangle = \sum_n \frac{1}{\langle \psi_n^+ \eta \psi_n \rangle} \left\langle \psi_n^+ \eta e^{-\beta\mathcal{H}} \psi_n \right\rangle = \\
&= \sum_n \frac{1}{\langle \hat{\psi}_n^+ \hat{\psi}_n \rangle} \left\langle \hat{\psi}_n^+ \underbrace{S^{-1}S}_{1} \underbrace{SS^{-1}}_{1} e^{-\beta\hat{\mathcal{H}}} \underbrace{SS^{-1}}_{1} \hat{\psi}_n \right\rangle = \\
&= \sum_n \left\langle \hat{\psi}_n^+ e^{-\beta\hat{\mathcal{H}}} \hat{\psi}_n \right\rangle = \hat{Z},
\end{aligned}
\tag{A9}
$$

where we have denoted the partition function of the Hermitian partner by $\hat{Z}$.

We can generalize (A8) to any product of field operators. Then, we obtain that:

$$\langle \Psi^+(0)\Psi(\tau')\rangle_{bi} = \langle \hat{\Psi}^+(0)\hat{\Psi}(\tau')\rangle =$$
$$= -\langle \hat{\Psi}^+(\beta)\hat{\Psi}(\tau')\rangle = -\langle \Psi(\beta)^+\Psi(\tau')\rangle_{bi}, \tag{A10}$$

so we conclude that the averages performed with the bi-orthogonal basis and with the density matrix $\rho = e^{-\beta\mathcal{H}}$ satisfy a KMS boundary condition, and thus, this state defines a thermal state in equilibrium, since it is trivial to modify the previous reasoning by including the Dyson time ordering operation. Furthermore, this reasoning justifies the definition of the discrete-frequency Green function $G_0(i\omega_n)$ in Equation (17) of the main text.

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
