# Peer review of "Non-Hermitian Chiral Magnetic Effect in Equilibrium"

_symmetry, doi:10.3390/sym12050761_

Round 1

Reviewer 1 Report

(see attached report)

Author Response

We are grateful to the Referees for raising the questions about our manuscript. We provide a point-by-point response to the referee comments in the attached PDF file. 

Reviewer 2 Report

In the manuscript “Non-Hermitian chiral magnetic effect in equilibrium”, authors discuss chiral magnetic effect (CME) in Chiral matter using the biorthogonal quantum mechanics formalism as the emergent gauge field with Weyl band and non-Hermitian PT symmetry. They showed how to break the statement of absence of the CME in equilibrium, concerning the existence of equilibrium currents in solids in the thermodynamic limit. The manuscript is well-organized and written for the quasi-Hermiticity so it is quite good for this journal.

The manuscript need minor corrections as following for keeping consistency and some questions for my eagerness to understand this theory. For the abbreviations or nomenclators, we need to write the explanation at least a single sentence.

Erratum in line 56, 181,188

What is the CSE (Chiral separation effects?) which is not defined in the line 63.

KMS in line 102 is not defined (in line 177).

Is “the eigenstates” “eigenvalues” in line 116 or keeping?

The index n is related to the Matsubara frequency from equation (11) but why the n is the Landau level index after equation (14).

The abbreviation, CSE, comes from the chiral current operator in line 130. Is it right or mistake?

What kind of system we can consider the non-Hermitian mass term m_5\gamma_5 in physical situation like equation(3)?

What is emergent field B_3 as external magnetic field for new Hamiltonian is there some definition from original non-Hermitian Hamiltonian?

Author Response

(The authors gave the same response as above.)

Reviewer 3 Report

Report on the manuscript entitled
"Non-Hermitian Chiral Magnetic Effect In Equilibrium"
by Chernodub and Cortijo

I have read with interest the manuscript entitled
"Non-Hermitian Chiral Magnetic Effect In Equilibrium".

I have three main criticisms to the manuscript.

1) The authors write a very nice introduction going
"into the middle of things". However, I believe that
such an introduction would be much better suited
to a more specialized journal. To me, given the large
audience of Symmetry, even if the paper has been sent
to the Special Issue "Relations between Condensed Matter
Physics and Relativistic Quantum Field Theory",
I would like the authors to write one more section in their
manuscript in order to explain the main concepts and jargon of
the specific research area of their study.
As a matter of fact, the topic of this special issue spans
many subfields and we cannot expect that every reader of one
subfield is also fluent in the other.

2) My second criticism is more conceptual since what the authors
claim to be doing got me a little bit confused. Basically, the
authors write that they are able to theoretically achieve the
chiral magnetic effect AT EQUILIBRIUM in a fermionic system by
using the formalism of non-Hermitian quantum mechanics.
Now, there is really a wide consensus that non-Hermitian quantum
mechanics in general represents system OUT OF EQUILIBRIUM [see
Rotter and Bird, A review of progress in the physics of open quantum
systems: theory and experiment, Rep. Prog. Phys. 78, 114001 (2015)].
In particular, Carl Bender, who is one of the "fathers" of PT-Symmetry,
has clearly explained in the first chapter of his recent book [C. M.
Bender, Pt Symmetry: In Quantum And Classical Physics (2018)] that
PT-symmetric systems are systems in a stationary NON-EQUILIBRIUM state.
Hence, I believe it is really necessary that the authors explain their
claim about describing an equilibrium non-Hermitian system.

3) The third criticism regards the fact that the the authors start
from Eqs. (4) and (5), defining the Heisenberg scheme of motion in a
non-Hermitian framework. Now, it should very well-known that the
Schrödinger and Heisenberg pictures are not equivalent in non-Hermitian
quantum mechanics, see [Graefe, Höning and Korsch, Classical limit of
non-Hermitian quantum dynamics—a generalized canonical structure,
J. Phys. A 43, 075306 (2010)] and [Sergi and Zloshchastiev, Non-Hermitian
Quantum Dynamics of a Two-level System and Models of Dissipative Environments,
Int J Mod Phys B 27, 1350163 (2013)]. The problem is that, within the
framework of non-Hermitian Quantum Mechanics, the density matrix must
be normalized in a time-dependent manner otherwise one cannot define a
proper probability and meaningful average properties cannot be calculated.
Hence, the questions are:
3a) How do the authors define a meaningful normalized probability starting
from Eqs. (4) and (5)?
4a) How do the authors define quantum averages starting from Eqs. (4) and (5)?
Definitely, my suggestion is to start their theory from the Schrödinger scheme
of motion.

Minor Points
- Line (26): perhaps "indices" should read "induces".
- Above Eq. (5), inside the parenthesis: "pleas not that ..."
should read "please note that ..."

I recommend a thorough spell-checking of the manuscript.

In conclusion, unless I receive a satisfactory answer to the
criticisms above, I cannot recommend the publication of
the manuscript in its present form.

Author Response

(The authors gave the same response as above.)

Round 2

Reviewer 1 Report

This work is now better presented, and is definitely interesting, I recommend its publication in Symmetry 

Reviewer 3 Report

There might be some minor typos in the manuscript. I found only one on line 213, p. 14, sez. Conclusions: "Another fact to pay attention is that there not an unique metric operator" should read "Another fact to pay attention is that there IS not an unique metric operator" ...

I think that things like these can be corrected in the proofs stage. But please do take them into accounts.